# Limited carry-over effects of socioemotional manipulations on subsequent unrelated memory tasks

Jaclyn H. Ford[1]*, Ryan Daley[2], Julia Maybury[3], Cortney Stedman[4], Julia Swiatek[1], Rachel Van Boxtel[5,6], Erin Welch[7], Elizabeth Kensinger[1]

**1** Department of Psychology and Neuroscience, Boston College, Chestnut Hill, MA, United States of America, **2** Department of Psychology, Gordon College, Wenham, MA, United States of America, **3** Department of Psychology, University of Cambridge, Cambridge, United Kingdom, **4** Cognitive Neuroscience Program, Florida International University, Miami, FL, United States of America, **5** Pacific Graduate School of Psychology, Palo Alto University, Palo Alto, CA, United States of America, **6** Department of Psychiatry and Behavioral Sciences, Stanford University School of Medicine, Stanford, CA, United States of America, **7** Department of Psychology, Columbia University, New York, NY, United States of America

\* JaclynHFord@gmail.com

**Data Availability Statement:** All data (4 files) are available from the OSF database (DOI 10.17605/OSF.IO/3P6MA).

## Abstract

Although age is typically associated with significant impairments in memory performance, several domains exist in which these impairments are reduced or even eliminated. These "pockets of preservation" in older adults' memory can be seen in tasks involving socioemotional processing and may be supported by distinct encoding or retrieval modes relative to neutral content. The current study examines whether engaging in socioemotional tasks *prior to* encoding or retrieval allows older adults to enter an encoding or retrieval mode that better supports memory performance. In two online studies, adults across the lifespan were asked to complete a memory task where they incidentally encoded and retrieved neutral (Experiment 1, *N* = 1621) or emotional (Experiment 2, *N* = 409) word-image pairs. Participants were randomly assigned to a control (i.e., no manipulation), pre-encoding, or pre-retrieval socioemotional manipulation condition. There were no main effects of manipulation condition, suggesting that such manipulations may not reliably enhance memory. However, future research is needed to follow up on exploratory analyses that highlighted particular conditions under which these manipulations may convey benefits. There were also no age-by-manipulation interactions. While these null effects may suggest that these manipulations are not better suited to older adults, this may also be a result of the unexpected *age-related increases* (Experiment 1) and *age invariance* (Experiment 2) in overall memory accuracy in the current study. Socioemotional manipulations should also be examined in older adults who underperform younger adults.

## Introduction

The world's population is aging rapidly [1], with individuals over the age of 65 holding a disproportionate percentage of positions of power in universities, major corporations, and

**Funding:** National Science Foundation (BCS-1823795) to EAK and JHF The funders had no role in study design, data collection and analysis, decision to publish, or preparation of the manuscript.

**Competing interests:** The authors have declared that no competing interests exist.

government [2], as well as making up a growing subset of our workforce [3]. Given these trends, it has become increasingly important to identify strategies that optimize cognition in this population. It is well-documented that healthy aging is associated with significant declines in certain areas of cognition, including episodic memory [4]. Contributing to these deficits is a difference in how young and older adults are able to mentally prepare themselves for a memory task: While young adults are able to maintain a "retrieval mode"–a brain state that can help them to activate relevant content and search processes [5]–older adults have difficulty initiating these proactive states, resulting in reduced memory vividness and detail [6]. Such difficulties may be driven by the fact that these proactive retrieval modes depend on neural networks and cognitive processes known to shown the greatest declines with increased age. The current study examines whether older adults' episodic memory may benefit from proactively activating mental states that pull on age-specific strengths, rather than attempting to utilize strategies that work well in young adults.

One often-reported strength in older adults is episodic memory for content with motivating, self-relevant, or emotional significance [7–9]. The relative preservation of memory for socioemotional information may help explain why older adults' memory impairments in daily life are often not as large as those seen in laboratory studies [10]. For example, older adults are often able to successfully retrieve autobiographical events [11, 12], as well as information that contributes to impression formation [13, 14] or item safety [15, 16]. Importantly, these "pockets of preservation" in older adults' memory are not reserved for items that intrinsically contain socioemotional significance. These memory enhancements can extend to otherwise neutral items that have undergone socioemotional manipulations at encoding or retrieval [8, 13, 15, 17, 18]. For instance, when older adults encode neutral words in a self-referential manner, their memory is better than when they asked to process these words in relation to another individual [19, 20].

Socioemotional processing may enhance encoding and retrieval in older adults because it allows the memory to be supported by different brain regions or encoding or retrieval modes than memory for neutral content. In particular, while memory for typical laboratory stimuli depends heavily on lateral prefrontal (lPFC) control processes [21, 22], memory in socioemotional domains relies more heavily on medial prefrontal (mPFC) control processes [8, 13, 23]. Older adults struggle to engage lPFC regions efficiently [24], making it difficult for them to encode new information into durable memory representations [25, 26] or bind new information into a novel event [27]. In contrast, the mPFC shows little age-related change in structure or function [28, 29] and can support older adults' successful encoding [13, 17, 19, 23] and retrieval [30, 31]. During socioemotional memory tasks, older adults may be able to compensate for deficits in lPFC recruitment by relying on networks that incorporate the mPFC [32, 33].

The research above shows that socioemotional processing that occurs *during* encoding or retrieval can help older adults overcome memory impairments. However, socioemotional processing occurring *prior to* a memory task may also be able to influence performance [34, 35]. Tambini and colleagues [35] demonstrated that, in young adults, neutral stimuli encoded up to a half hour after an emotional memory encoding task were better remembered than neutral stimuli that were not preceded by an emotional memory task. However, it is unclear whether older adults would benefit from the carry-over effects of emotional processing in the same way. One study [34] showed that older adults' autobiographical memory detail could be enhanced by having them provide self-statements prior to retrieval, but this enhancement has not yet been shown for an unrelated, non-socioemotional task.

The current study tests the possibility that the carry-over effects of socioemotional processing may introduce an optimal encoding or retrieval mode for older adults. In two episodic

memory studies, participants were randomly assigned to one of three manipulation conditions: *control* (participants do not complete a socioemotional task prior to encoding or retrieval), *pre-encoding* (participants complete a socioemotional task prior to encoding), or *pre-retrieval* (participants complete a socioemotional task prior to retrieval). We employed three socioemotional tasks (*self-reference statements*, *listening to music*, and *autobiographical memory retrieval)* that are known to recruit the mPFC in individuals across the lifespan [19, 36, 37] and, in prior studies, have been shown to enhance memory detail when engaged *during* encoding and retrieval [8, 38–40], or in the case of "Who am I?" reflections, to prospectively enhance autobiographical memory in older adults [34].

*Experiment 1* extends prior research [35] by testing the hypothesis that introducing a socioemotional manipulation prior to encoding or retrieval of *neutral* word-image pairs will enhance older adults' memory accuracy. Increased memory for those in the pre-encoding or pre-retrieval manipulation conditions would suggest that socioemotional processing is not only beneficial to older adults' memory processes *in the moment*, but can also induce an encoding or retrieval mode that optimizes subsequent memory tasks. Although the lack of a significant benefit across all older adults might argue against such an induced encoding or retrieval mode, it is also likely that socioemotional manipulations may be particularly beneficial for certain participants (e.g., those who are less able to rely on traditional learning strategies) or in certain circumstances (e.g., when instructions are presented at a particular timepoint during the task). Therefore, exploratory follow-up analyses will examine whether there are individual or situational differences in benefits of the manipulations, even in the absence of a main effect.

Additionally, because older adults already exhibit relative preservation for content that intrinsically contains social or emotional content, *Experiment 2* tests the mnemonic benefits of introducing a socioemotional manipulation prior to encoding or retrieval of *emotional* word-image pairs. Because older adults tend to remember more positive relative to negative information (i.e., the *positivity effect* [41]), a final set of analyses tests the interaction of image valence (positive v. negative) with manipulation condition.

## Experiment 1

### Materials and methods

**Participants.** Data in Experiment 1 come from 1,753 online participants (ages 18–94, $M = 49.52$, $SD = 19.76$, 58% female, 51% with a college degree or more, 81% white and 92% not Hispanic) collected between 5/23/19 and 4/30/20. Participants all passed attention and quality assurance checks. 132 participants were excluded for having memory performance below chance, so the final analyses included data from 1,621 participants (ages 18–94, $M = 50.62$, $SD = 19.58$, 58% female, 52% with a college degree or more, 84% white and 93% not Hispanic). Power analyses indicate that with this sample size, we had 89% power to detect even a small difference between manipulations ($\eta^2_p = .01$). Participants provided written consent in accordance with the requirements of the Institutional Review Board at Boston College. Participants recruited using Amazon Mechanical Turk were compensated $14; participants recruited through Qualtrics were paid via their agreements with Qualtrics.

After consenting and providing demographic information, participants were randomly assigned to one of six manipulation conditions: No manipulation (Control, $N = 279$), preencoding self-statement task (Encoding-Self, $N = 261$), pre-encoding music task (Encoding-Music, $N = 284$), pre-retrieval self-statement task (Retrieval-Self, $N = 260$), pre-retrieval music task (Retrieval-Music, $N = 250$), or pre-retrieval autobiographical memory task (Retrieval-Autobio, $N = 287$). See *Socioemotional manipulations* section below for a description of the

**Table 1. Demographic information as a function of experiment and condition.**

| | | Age | | | % Female | % College degree |
|---|---|---|---|---|---|---|
| | N | Min | Max | Average (SD) | | |
| *Experiment 1 (Neutral Images)* | | | | | | |
| Control | 279 | 18 | 81 | 52.00 (18.75) | 63% | 46% |
| Encoding-Music | 284 | 18 | 87 | 51.43 (19.94) | 53% | 46% |
| Encoding-Self | 261 | 18 | 94 | 49.73 (19.94) | 57% | 57% |
| Retrieval-Autobio | 287 | 18 | 86 | 50.92 (19.39) | 63% | 53% |
| Retrieval-Music | 250 | 18 | 86 | 49.78 (19.50) | 60% | 52% |
| Retrieval-Self | 260 | 18 | 83 | 49.66 (20.01) | 55% | 57% |
| Total | 1621 | 18 | 94 | 50.62 (19.58) | 58% | 53% |
| *Experiment 2 (Emotional Images)* | | | | | | |
| Control | 68 | 18 | 74 | 48.47 (21.43) | 62% | 38% |
| Encoding-Music | 63 | 18 | 77 | 47.76 (21.18) | 64% | 48% |
| Encoding-Self | 71 | 18 | 76 | 51.07 (21.44) | 70% | 45% |
| Retrieval-Autobio | 65 | 18 | 79 | 47.23 (21.72) | 74% | 35% |
| Retrieval-Music | 69 | 18 | 79 | 49.16 (21.43) | 67% | 38% |
| Retrieval-Self | 73 | 18 | 79 | 53.18 (21.20) | 69% | 47% |
| Total | 409 | 18 | 79 | 49.57 (21.37) | 66% | 43% |

three socioemotional manipulations. See Table 1 and S1 and S2 Tables for more details about the demographic breakdown across conditions and S1 Fig for age distributions by condition, but conditions did not differ as a function of age ($F(1,5) = .71$, $p = .62$, $\eta^2_p = .002$), education ($\chi^2(25) = 37.12$, $p = .06$), ethnicity ($\chi^2(5) = 2.10$, $p = .84$), or race ($\chi^2(25) = 17.01$, $p = .88$). Participant sex assigned at birth did differ across conditions, with the Control and Retrieval-Autobio groups having a greater proportion of females compared to the other groups ($\chi^2(5) = 11.31$, $p = .046$). For primary analyses, all individuals who completed a socioemotional task prior to encoding were combined into a *Pre-Encoding Manipulation* group and all individuals who completed a socioemotional task prior to retrieval were combined into a *Pre-Retrieval Manipulation* group.

**Procedure.** See Fig 1 for visual depiction of task protocol. First, participants in the *pre-encoding manipulation* condition completed either the *Self-Reference* or the *Music* task and those in the *control* and *pre-retrieval* conditions completed the Test of Everyday Attention (TEA; Robertson et al., 1996) as a simple attention control task. These tasks were immediately followed by a short incidental encoding task in which participants were presented with 80 neutral word-image pairs and were asked to judge the appropriateness of each word as a description of the image (Fig 1). Participants had 4 seconds to make each judgement, but could advance the screen at any time after providing a response.

Following encoding, there was a 20-minute delay during which participants completed the Pittsburgh Sleep Quality Index (PSQI [42]), Physical Activity Scale for the Elderly (PASE [43]), 36-Item Short Form Health Survey (SF-36 [44]), an n-back working memory task [45, 46], and the Community Integration Questionnaire (CIQ [47]). Participants in the *pre-retrieval manipulation* condition then completed the *Self-Reference*, *Music*, or *Autobiographical Memory* socioemotional task, while those in the *control* and *pre-encoding* conditions completed the Shipley vocabulary test [48].

These tasks were followed by a retrieval task in which participants were presented with 160 neutral words, half of which had been studied during encoding. Participants responded with a "0" if the word was new (i.e., not studied), or a 1–5 if the word was old (i.e., studied). If old, the

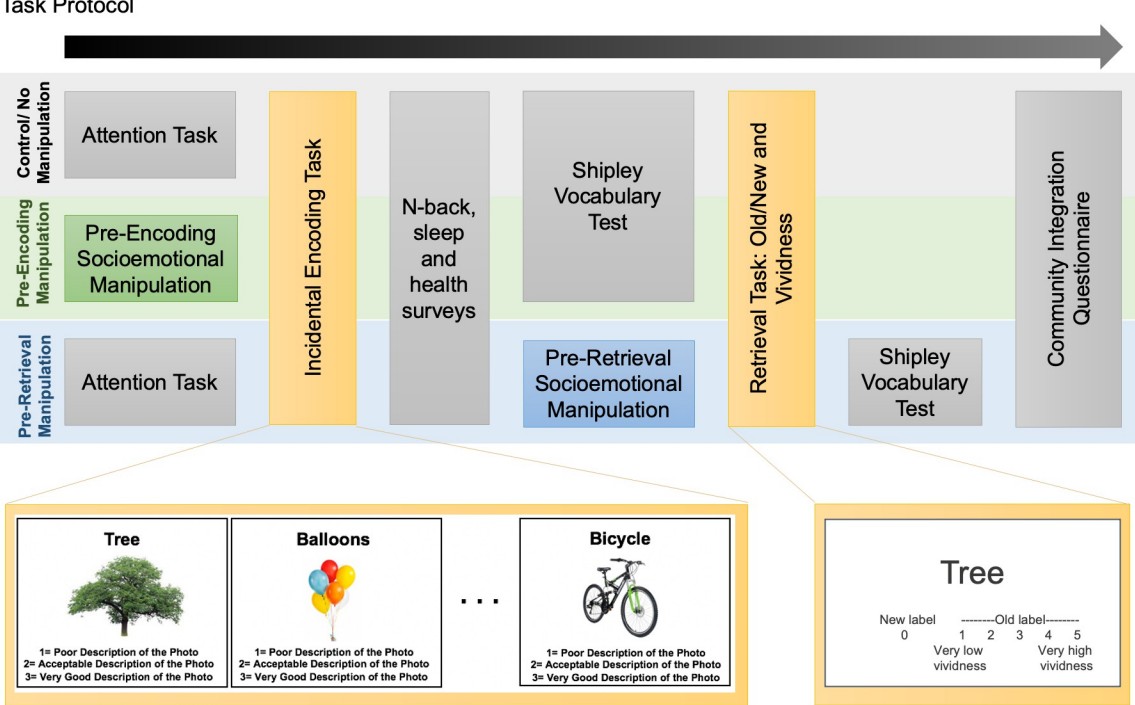

**Fig 1. Visual depiction of the experimental protocols used in Experiments 1 and 2, including example slides from the incidental encoding task and retrieval task.**

participants used the 1–5 scale to report how vividly they remembered encoding the word-image pair (Fig 1). As with encoding, participants were given 4 seconds to respond, but could advance the screen whenever ready. Participants in the *pre-retrieval* condition completed the Shipley vocabulary test [48] after retrieval, then the final task for all participants was CIQ [47].

**Socioemotional manipulations.** *Self-reference statements (Encoding-Self or Retrieval-Self conditions).* Reflecting on "Who am I?" prior to retrieval has been shown to enhance autobiographical memory in older adults [34]. The current study adapts this methodology by placing the self-reference statement task immediately prior to an episodic memory *encoding* or *retrieval* task. Participants in the *Self-Reference* manipulation conditions were provided with the following instructions:

*Please produce 20 statements answering the question*

*"Who am I?"*

*Each statement should begin with "I am..." Please respond to this with statements that reflect stable and enduring aspects of your identity; these might include personality traits, physical traits, relationships, occupations, talents, etc.*

*You will have 5 minutes to complete this task. The screen will automatically progress after the 5 minutes are over.*

For individual difference analyses, each statement was rated as being either positive or negative by two independent researchers. A subset of 40 statements were scored by both researchers to confirm reliability (cronbach's $\alpha$ = .99, with inconsistent responses discussed to agreement), and the remaining were divided between researchers.

*Listening to music (Encoding-Music or Retrieval-Music).* Participants in the *Music* condition were presented with a 5-minute piece of pleasant classical music (Piano Concerto No. 21 in C, K. 467: III. Allegro Vivace assai). A pleasant piece of music was selected, as pleasantness has been shown to track with mPFC recruitment [49–52]. There was not a task related to the music, so participants were just provided with following instructions:

Please listen to the music. Please do not complete other activities while the music plays.

*Autobiographical memory retrieval (Retrieval-Autobio).* Participants in the *Autobiographical Memory* condition were provided with the following instructions:

*Please recall a specific event that occurred in the past year, but not in the past week. This should be an event that lasted no longer than 24 hours and was specific to a particular time and place.*

*Describe this event in as much detail as possible, including what you remember about what you saw, smelled, heard, and felt, as well as anything you remembering thinking at the time. Please provide the entire sequence of events.*

*You will have 5 minutes to complete this task. The screen will automatically progress after the 5 minutes are over.*

It is important to note that the *Autobiographical Memory* task was only included as a *pre-retrieval* manipulation, as we did not want to bias participants to expect a memory task prior to the incidental encoding task by having them complete a memory task immediately before viewing the images.

For individual difference analyses, all memory narratives were analyzed for positive and negative word use using the Linguistic Inquiry and Word Count software [53]. To obtain a score of narrative specificity (i.e., the extent to which a memory was specific to a particular place and time), narratives were scored by two independent researchers using the Autobiographical Interview [54]. In the Autobiographical Interview, researchers divide each narrative into individual details and assign each as "internal" (i.e., related to the central event in the narrative) or "external" to this event (e.g., background semantic content or related to another event). A subset of 30 narratives were scored by both researchers to confirm reliability (cronbach's α = .84, with inconsistent responses discussed to agreement), and the remaining were divided between researchers.

**Additional measures.** *Test of Everyday Attention (TEA [55]).* For the current study, we utilized an adapted version of the TEA in order to accommodate our online survey. Participants were presented with a list of 130 "phone book listings" for local hotels. Each listing included a hotel name, a pair of symbols, and a phone number (e.g., *Adams, The Hotel. . .O% 328–4711, Allons Hotel. . .OX 435–1911*, etc). Participants were told that hotels with two of the same symbol (e.g., *Ancho Inn. . .XX 584–9265*) were preferred hotels, so they should select those from the list. Participants' TEA scores were calculated by subtracting the proportion of False Alarms (i.e., incorrectly selecting a listing with two different symbols) from the proportion of Hits (i.e., correctly selecting a listing with two of the same symbol). Because this was used as a control task prior to encoding, participants who were in the pre-encoding manipulation conditions do not have data for this test.

*Pittsburgh Sleep Quality Index (PSQI [42]).* The PSQI consisted of 11 questions that asked participants about the *quantity* and *quality* of their sleep during the past month. For the current study, we focused on sleep *quantity*, only, and only looked at how memory related to the number of hours of sleep the participant reported getting on average.

*n-back working memory task [45].* To measure working memory ability, the current study asked participants to complete a 2-back task. In this task, participants were presented with a series of letters and instructed to respond "yes" whenever a presented letter was the same as the letter *two before.* This task requires the participant to update and maintain a stimulus and inhibit distractors [56]. In order to control for individual differences in attention and processing speed, the current study also included a 0-back, where participants were again presented with a series of letters, but were now just instructed to respond for a particular letter (e.g., "respond *yes* when you see the letter *k*"). Accuracy for both tasks was calculated as the proportion of correct responses to target letters, minus the proportion of incorrect responses to non-targets. The difference between 0-back accuracy and 2-back accuracy reflects reductions in performance due to the working memory load, with greater values reflecting larger reductions due to working memory.

*Shipley vocabulary test [48].* In the Shipley vocabulary test, participants are presented with 40 words and are asked to identify a synonym for each out of four options. A participant's score, out of a possible 40, is taken as a measure of crystallized knowledge.

*Community Integration Questionnaire (CIQ [47]).* The CIQ asked participants questions about their involvement in activities in the home (*home integration*), with others (*social integration*), and related to work or school (*productive activities*). Larger scores reflect greater involvement and integration with the community. Although the three sub-scores can be used separately, the current study only uses the total score as a covariate.

*Physical Activity Scale for the Elderly (PASE [43]).* The PASE is a short survey that can be used to measure physical activity for the prior week. It uses reported frequency, duration, and intensity level of various activities (e.g., walking, housework, exercise) to provide an overall activity score. This measure was included in the survey, but was not included in the exploratory analyses reported here.

*36-Item short form health survey (SF-36 [44]).* The SF-36 is a 36-item survey that asks participants questions about the overall quality of their health and well-being. Questions focus on how participants have felt over the past four weeks, as well as how they feel in comparison to a year prior. As with the PASE, the measure was included in the survey, but was not included in the exploratory analyses reported here.

*COVID timing.* A vast majority (*N* = 1401, 86.4%) of the participants in Experiment 1 were tested prior to the COVID-19 pandemic and associated shut-downs. However, the remaining 220 participants (13.6%) were tested immediately after shut-downs had been announced. Because these changes may have contributed to shifts in memory and emotional state, exploratory analyses were conducted with COVID timing as an additional variable.

**Data analysis.**   *Memory accuracy* for the retrieval task was calculated by subtracting each participant's *false alarm rate* (i.e., the proportion of *incorrect* "old" responses to new items) from their *hit rate* (i.e., the proportion of *correct* "old" responses to old items). We conducted a Factorial ANCOVA with *manipulation* (pre-encoding, pre-retrieval, and none/control) as a between-subjects factor and *age* (treated as a continuous variable) as a covariate of interest. Although the current study focused on accuracy ratings, vividness ratings were also collected. See S3 File for ANCOVA looking at effects of condition and age on average vividness ratings.

Although the primary goal of the current study was to determine the efficacy of a socioemotional manipulation, follow-up analyses were conducted separately for each manipulation, separately. These analyses compared each manipulation directly to the control condition and, in the case of the *self* and *music* manipulations, compared the efficacy of the manipulation at two different time points (e.g., pre-encoding self v. pre-retrieval self). In addition, analyses conducted for the *autobiographical memory* and *self* manipulations examined potential effects of how the participant completed the task. For instance, we examined the role of

autobiographical memory specificity and valence in determining how the manipulation differed from control. Similarly, we examined how the valence of self-statements influenced memory.

Finally, the online nature of the study meant that there was considerable variability in our sample. Exploratory analyses examined the potential effects of individual differences in working memory ability, attention, crystallized knowledge, sleep, and community involvement (see above to descriptions of each measure). For each analysis, the measure of interest was added as a second covariate of interest. An additional exploratory analysis compared participants who had been tested prior to stay-at-home orders associated with the COVID-19 pandemic (here, operationalized as March 10, 2020), and those tested after. In this analysis, COVID timing was added as an additional between-subjects factor.

## Results

Surprisingly, age was associated with increased accuracy ($F(1,1615) = 55.70$, $p < .001$, $\eta^2_p = .03$; $r = .19$, $p < .001$; See Fig 2A), and this effect of age did not differ across manipulation conditions ($F(1,1615) = 1.27$, $p = .28$, $\eta^2_p = .002$). There was a trending main effect of manipulation condition ($F(1,1615) = 2.81$, $p = .06$, $\eta^2_p = .003$), driven by significantly greater accuracy in individuals who participated in a memory manipulation prior to retrieval ($M = .68$, $SE = .007$) compared to prior to encoding ($M = .65$, $SE = .009$).

Exploratory analyses were conducted to examine individual manipulation conditions. The only significant difference ($F(1,807) = 5.755$, $p = .003$, $\eta^2_p = .02$) was between people who participated in the music manipulation prior to retrieval ($M = .70$, $SE = .01$) and those who participated the same manipulation prior to encoding ($M = .64$, $SE = .01$, $p = .001$; See Fig 3A). Neither differed significantly from Control ($M = .67$, $SE = .01$). Once again, age was associated with increased accuracy ($F(1,807) = 34.45$, $p < .001$, $\eta^2_p = .04$), but this did not differ across conditions ($F(1,807) = 1.59$, $p = .22$, $\eta^2_p = .004$).

No other differences were identified. Specifically, memory accuracy did not differ between the Autobiographical memory manipulation condition and the Control condition ($F(1,562) = .005$, $p = .95$, $\eta^2_p < .001$). Age was associated with increased accuracy ($F(1,562) = 17.95$, $p < .001$, $\eta^2_p = .03$), but this did not differ between conditions ($F(1,562) = 1.62$, $p = .20$, $\eta^2_p = .003$). Similarly, memory accuracy did not differ across control, pre-encoding *self*, and pre-retrieval

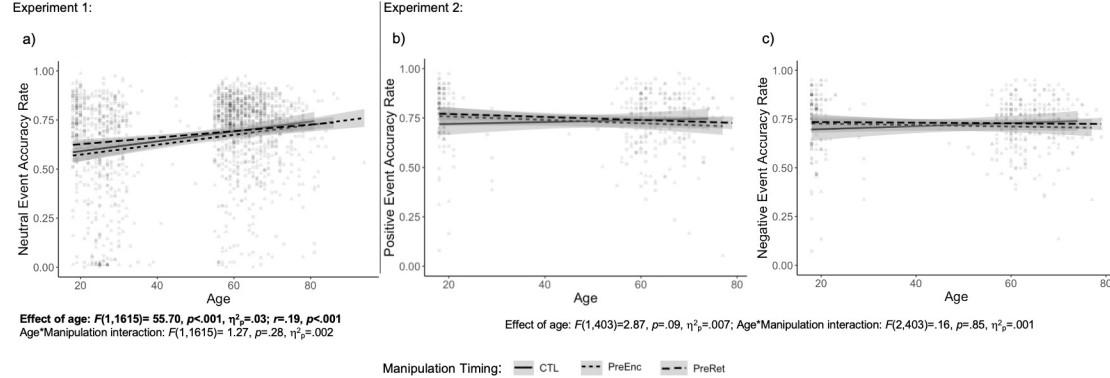

**Fig 2.** Relation between age and memory accuracy as a function of socioemotional manipulation timing for neutral events (a, Experiment 1) and positive (b) and negative (c) events (Experiment 2). The control condition is depicted by a solid line, pre-encoding manipulations are depicted by a short-dash line, and pre-retrieval manipulations are depicted by a long-dash line. Age is associated with a significant memory increase for neutral events (a), but not positive (b) or negative (c). Age does not interact with manipulation timing for any condition.

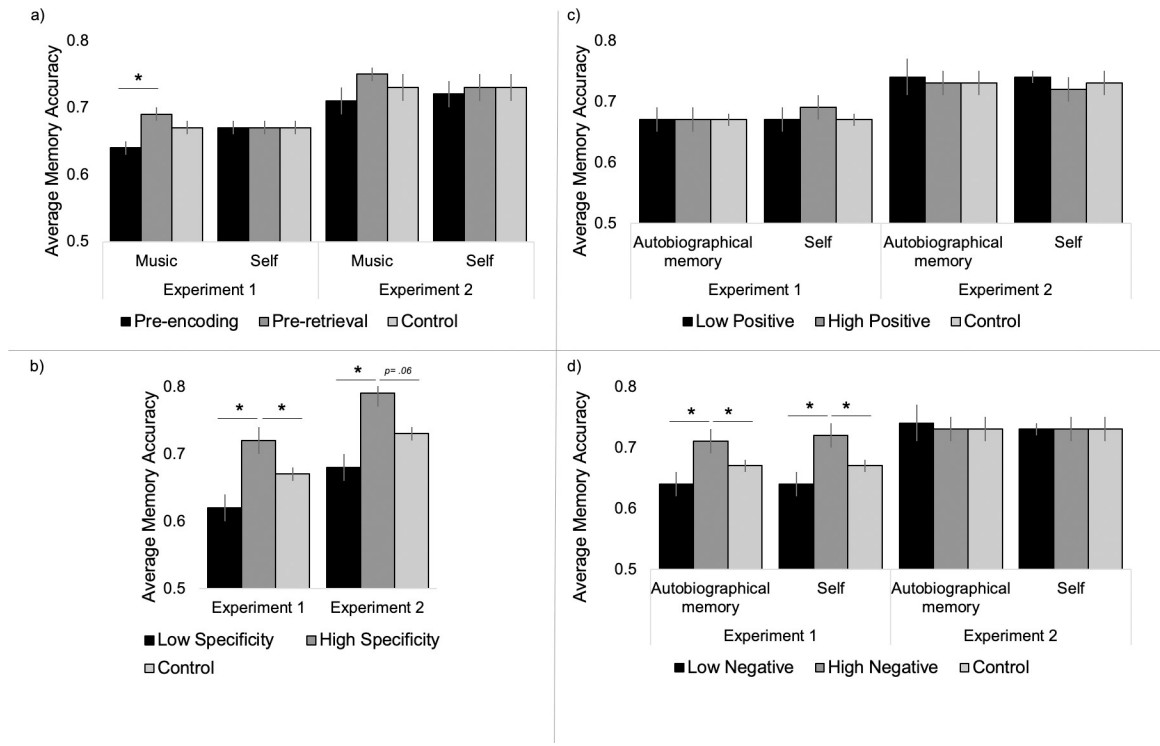

**Fig 3. Effects of individual differences on manipulation effects on memory accuracy.** A) Timing significantly influenced the effect of music in Experiment 1, only. B) Individuals who retrieved more specific autobiographical memories had better episodic memory accuracy than those who did not retrieval autobiographical memories (controls) and those who retrieved less specific autobiographical memories in both Experiments. C) Autobiographical memory and self-statement positivity were not related to episodic memory in either Experiment. D) Individuals who retrieved more negative autobiographical memories and individuals who retrieved more negative self statements had better episodic memory accuracy than those who did not retrieve either an autobiographical memory or self-statements and those who retrieved less negative information in Experiment 1, but not in Experiment 2.

*self* conditions ($F(1,794) = .04$, $p = .96$, $\eta^2_p < .001$). As before, age was associated with increased accuracy ($F(1, 794) = 35.19$, $p < .001$, $\eta^2_p = .04$), but this did not differ between groups ($F(1, 794) = .10$, $p = .91$, $\eta^2_p < .001$). These same patterns persisted when removing the 91 participants from the Self Statement manipulation who did not provide the full 20 statements (age: $F(1, 703) = 34.70$, $p < .001$, $\eta^2_p = .05$; group: $F(1, 703) = .97$, $p = .38$, $\eta^2_p = .003$; age-by-group interaction: $F(1, 703) = .41$, $p = .66$, $\eta^2_p < .001$).

**Individual difference analyses.** By considering different ways in which participants may engage with the autobiographical memory task, we identified some potential benefits of the manipulation for subsequent episodic memory. First, we compared participants in the control group to individuals in the autobiographical memory condition with *highly specific* autobiographical memories ($N = 114$) and to those with *less specific* autobiographical memories ($N = 127$). There was a significant effect of group ($F(1,560) = 7.38$, $p = .001$, $\eta^2_p = .03$), where participants with more specific memories ($M = .72$, $SE = .02$) had greater memory accuracy than those with less specific memories ($M = .62$, $SE = .02$, $p < .001$) and those in the control group ($M = .67$, $SE = .01$, $p = .01$; See Fig 3B), which did not differ from one another ($p = .06$). Because age was associated with internal detail ($r = -.25$, $p < .001$), we controlled for age in this model. Age was associated with increased accuracy ($F(1,560) = 16.97$, $p < .001$, $\eta^2_p = .03$) and this did not differ across conditions ($F(1,560) = 2.00$, $p = .14$, $\eta^2_p = .007$).

We also looked at how the valence of the autobiographical memory retrieved during the autobiographical memory manipulation influenced the effects. First, we compared participants

in the control condition to those in the autobiographical memory condition who used a high proportion of positive emotional words in their narratives ($N$ = 120), and those with a low proportion of positive words ($N$ = 121). There was no effect of positive word use on episodic memory performance ($F(1,560)$ = .89, $p$ = .41, $\eta^2_p$ = .003; See Fig 3C) and the significant effect of age ($F(1,560)$ = 12.24, $p$ = .001, $\eta^2_p$ = .02) did not interact with word use ($F(1,560)$ = 2.25, $p$ = .11, $\eta^2_p$ = .008). Similarly, we split participants in the Autobiographical Memory condition based on their negative word use ($N$ high = 117, $N$ low = 124). Negative word use was associated with increased episodic memory accuracy ($F(1,560)$ = 5.93, $p$ = .003, $\eta^2_p$ = .02), with greater memory accuracy for individuals who used more negative words ($M$ = .72, $SE$ = .02) than those using fewer negative words ($M$ = .63, $SE$ = .02, $p$ = .001) and those in the Control condition ($M$ = .67, $SE$ = .01, $p$ = .01; See Fig 3D). The significant effect of age ($F(1,560)$ = 13.07, $p$ < .001, $\eta^2_p$ = .02) did not differ across groups ($F(1,560)$ = 1.15, $p$ = .32, $\eta^2_p$ = .004).

The analyses described above revealed a benefit of retrieving an autobiographical memory prior to episodic memory retrieval, *but only if* the memory was negative and highly specific. Of note, autobiographical memory negativity and specificity were not related in this sample ($r$ = .08, $p$ = .24), and when both were included in the same model, there was a significant effect of both negativity ($F(1,558)$ = 8.63, $p$ = .003, $\eta^2_p$ = .02) and specificity ($F(1,558)$ = 12.06, $p$ = .001, $\eta^2_p$ = .02) on episodic memory accuracy, suggesting independent effects on memory.

The next set of analyses explore effects of the self statement manipulation. Because timing of the self statement manipulation did not influence memory, individual difference analyses were collapsed across pre-encoding and pre-retrieval groups. Similar to the analysis for the Autobiographical Memory manipulation analysis, we were interested in how the valence of self-statements during the self manipulation influenced memory accuracy. First, we compared participants who provided a greater number of positive self-statements ($N$ = 205) to those who provided fewer positive statements ($N$ = 225). There was no effect of positive statements on episodic memory performance ($F(1,703)$ = 1.27, $p$ = .28, $\eta^2_p$ = .004), and the significant effect of age ($F(1,703)$ = 35.47, $p$ < .001, $\eta^2_p$ = .05) did not interact with use of positive statements ($F(1,703)$ = .85, $p$ = .43, $\eta^2_p$ = .002; See Fig 3C).

We also split participants in the Self Statement condition based on the number of negative self-statements ($N$ high = 201, $N$ low = 229). Negative statement use was associated with increased episodic memory accuracy ($F(1,703)$ = 6.12, $p$ = .002, $\eta^2_p$ = .02), with greater memory accuracy for individuals who provided more negative statements ($M$ = .72, $SE$ = .01) than those providing fewer negative statements ($M$ = .66, $SE$ = .01, $p$ = .002) and those in the Control condition ($M$ = .67, $SE$ = .01, $p$ = .005). The significant effect of age ($F(1,703)$ = 29.82, $p$ < .001, $\eta^2_p$ = .04) did not differ across groups ($F(1,703)$ = 2.50, $p$ = .08, $\eta^2_p$ = .007; See Fig 3D). Exploratory analyses confirmed that the enhancing effect of negative self-statements was significant when the manipulation was conducted prior to encoding ($F(1, 213)$ = 4.78, $p$ = .03, $\eta^2_p$ = .01), and prior to retrieval ($F(1, 209)$ = 6.70, $p$ = .01, $\eta^2_p$ = .03). Similar to what was seen in the autobiographical memory analysis, these findings suggest a benefit of generating self statements prior to episodic memory encoding and retrieval, *but only if* the statements are negative.

Individual differences in COVID timing, attention, working memory, community integration, and sleep did not interact with the effect of manipulation on memory accuracy (see S1 File for these analyses). When controlling for vocabulary, there was a main effect of vocabulary ability ($F(1,1609)$ = 234.33, $p$ < .001, $\eta^2_p$ = .13), age ($F(1,1609)$ = 9.78, $p$ < .001, $\eta^2_p$ = .02), and manipulation ($F(1,1609)$ = 20.19, $p$ < .001, $\eta^2_p$ = .02), qualified by an age-by-vocabulary interaction ($F(1,1609)$ = 29.97, $p$ < .001, $\eta^2_p$ = .02), a vocab-by-manipulation interaction ($F(1,1609)$ = 36.29, $p$ < .001, $\eta^2_p$ = .04), and an age-by-vocab-by-manipulation interaction ($F(1,1609)$ = 3.82, $p$ = .02, $\eta^2_p$ = .005). The age-by-manipulation interaction was not significant ($F(1,1609)$ =

2.21, $p = .11$, $\eta^2_p = .003$).The three-way interaction was driven by a significant vocab-by-manipulation interaction in older ($F(1,763) = 14.73$, $p < .001$, $\eta^2_p = .04$), but not young adults ($F(1,543) = 1.91$, $p = .15$, $\eta^2_p = .007$). In older adults, higher vocabulary scores were associated with greater accuracy in participants who were in a pre-encoding manipulation condition ($r = .37$, $p < .001$) compared to those in a pre-retrieval manipulation condition ($r = .14$, $p = .008$) or the control condition ($r = .22$, $p = .01$). Although this may be a real effect of manipulation, it could also be a confound of lower overall vocabulary scores in the pre-retrieval manipulation ($M = 30.08$, $SE = .29$) relative to control ($M = 33.56$, $SE = .50$) and pre-encoding ($M = 33.07$, $SE = .35$; $F(2,1619) = 29.76$, $p < .001$, $\eta^2_p = .04$).

**Summary.** The primary analysis revealed a trend toward increased efficacy of socioemotional manipulations completed prior to *retrieval* relative to *encoding*. Follow-up analyses within individual manipulations revealed that this trend was being driven by a significant benefit for the pre-retrieval music manipulation relative to pre-encoding music. This same difference was not present for the self manipulation, suggesting that the temporal benefit may be specific to music.

Individual difference analyses also highlight the important fact that the benefit of socioemotional manipulations may differ depending on individual circumstances. In both the autobiographical memory and self conditions, individuals who provided negative content had better memory than those who did not *and* better memory than those in the control condition. Further, individuals who recalled more specific autobiographical events had better memory than both control participants and individuals with low specificity. There was no effect of manipulation timing on these effects. When controlling for performance on the Shipley vocabulary test, pre-encoding manipulations provided more of a benefit for older adults with greater crystallized knowledge, suggesting that socioemotional manipulations may allow older adults with more vocabulary ability to elaborate on items during encoding more than they would in the control condition.

Experiment 1 examined the benefits of participating in a socioemotional task prior to encoding or retrieval of neutral information. The results did not strongly support these benefits, although they suggest that socioemotional manipulations may be beneficial for *some participants* in *some circumstances*. Experiment 2 extends this research by examining how participating in a socioemotional task might influence subsequent encoding or retrieval of *emotional* information, looking at the difference between the effects on positive and negative images.

## Experiment 2

### Materials and methods

**Participants.** Data in Experiment 2 come from 417 participants (ages 18–79, $M = 49.14$, $SD = 21.40$, 66% female, 43% with a college degree or more, 81% white and 94% not Hispanic) tested between 7/23/20 and 12/7/20. None of these participants participated in Experiment 1. Eight of these participants were excluded for having memory performance below chance, so the final analyses included data from 409 participants (ages 18–79, $M = 49.57$, $SD = 21.37$, 68% female, 42% with a college degree or more, 81% white and 94% not Hispanic). Participants were randomly assigned to one of six memory conditions: No manipulation (Control, $N = 68$), pre-encoding self-statement task (Encoding-Self, $N = 71$), pre-encoding music task (Encoding-Music, $N = 63$), pre-retrieval self-statement task (Retrieval-Self, $N = 73$), pre-retrieval music task (Retrieval-Music, $N = 69$), or pre-retrieval autobiographical memory task (Retrieval-Autobio, $N = 65$). See Table 1 and S1 and S2 Tables for more information about demographic breakdown across conditions and S1 Fig for age distributions by condition, but

conditions did not differ as a function of age ($F(1,5) = .77$, $p = .57$, $\eta^2_p = .009$), sex ($\chi^2(5) = 3.00$, $p = .70$), education ($\chi^2(20) = 19.15$, $p = .51$), ethnicity ($\chi^2(5) = 3.43$, $p = .63$), or race ($\chi^2(20) = 27.47$, $p = .12$). Power analyses indicate that with this sample size, we had 42% power to detect a small difference between manipulations ($\eta^2_p = .01$), 99% power to detect a medium difference ($\eta^2_p = .06$), and 90% power to detect the age effect seen in Experiment 1. Participants provided written consent in accordance with the requirements of the Institutional Review Board at Boston College.

For analyses examining the differential effects of memory condition on neutral and emotional images, data from these participants were compared to those from Experiment 1. Age ($F(1,2028) = .91$, $p = .34$, $\eta^2_p < .001$) and ethnicity ($\chi^2(1) = .87$, $p = .35$) did not differ as a function of Experiment, but Experiment 1 had significantly fewer females ($\chi^2(1) = 11.05$, $p = .001$), more participants with a college degree or more ($\chi^2(5) = 13.67$, $p = .02$), and more Asian participants ($\chi^2(5) = 12.79$, $p = .03$).

An additional 351 participants from Experiment 1 also completed the Experiment 2 survey. See S4 File for within-subject comparisons across experiments.

**Procedure.** The procedure for Experiment 2 (Fig 1) was nearly identical to that of Experiment 1. The primary difference between experiments was that word-image pairs in Experiment 2 contained a neutral word paired with a positive or negative image, rather than a neutral image. In addition, because of the emotional content of the stimuli used in this experiment and the testing that was happening during the COVID pandemic, participants in Experiment 2 were asked to complete three additional affective measures at the end of the study:

*Geriatric Depression Scale (GDS).* The current study used a virtual version of the GDS-15 [57] in which participants responded to 15 yes-or-no questions about how they had felt during the prior week. Scores of 0–4 are considered normal, 5–8 indicate mild depression, 9–11 indicate moderate depression, and 12–15 indicate severe depression.

*Beck Depression Inventory (BDI).* The BDI [58] is made up of 21 questions that are self-rated on a scale of 0–3 and cover the different aspects of depression. In this scale, scores of 0–9 indicate minimal depression, 10–18 indicate mild depression, 19–29 indicate moderate depression, and 30–63 indicate severe depression.

*UCLA Loneliness Scale.* The current study uses the third version of the UCLA Loneliness Scale [59]. This version is made up of twenty questions that investigate both feelings of loneliness and feelings of social isolation. Answers are rated on a scale of 1 (Never) to 4 (Often), with half of the questions reversed scored. Scores range from 20 to 80, with scores of 20–34 reflecting low loneliness, 35–49 a moderate degree of loneliness, 50–64 a moderately high degree of loneliness, and 65–80 a high degree of loneliness.

**Data analysis.** As with Experiment 1, *memory accuracy* was calculated by subtracting each participant's *false alarm rate* (i.e., the proportion of *incorrect* "old" responses to new items) from their *hit rate* (i.e., the proportion of *correct* "old" responses to old items). We conducted a Factorial ANCOVA with *manipulation* (pre-encoding, pre-retrieval, and none/control) as a between-subjects factor, *valence* (positive v. negative) as a within-subjects factor, and *age* (treated as a continuous variable) as a covariate of interest. See S3 File for ANCOVA looking at effects of condition, valence, and age on average vividness ratings.

As before, follow-up analyses examined manipulation tasks, separately, as well as the individual difference effects of working memory ability, attention, crystallized knowledge, sleep, and community involvement (see above to descriptions of each measure). Analyses were conducted the same as in Experiment 1, but with valence included as an additional factor. COVID timing was not included as an individual difference in Experiment 2, as all participants were tested after the onset of the COVID-19 pandemic.

Experiment 1 v. Experiment 2 comparisons

A follow-up analysis was conducted to test whether the effect of a memory manipulation condition differed depending on the emotional content of the items in the memory task. For these analyses, positive and negative images in Experiment 2 were averaged together to produce a single *emotional images* measure. Specifically, this analysis compared the effect of a memory manipulation on memory accuracy and vividness for *neutral images* (Experiment 1) and for *emotional images* (Experiment 2), using an ANCOVA with Experiment (Experiment 1 v. Experiment 2) and Manipulation (Pre-encoding manipulation, Pre-retrieval manipulation, or Control) as between subject factors, and age as a continuous covariate.

## Results

Participants had better memory for positive ($M = .74$, $SE = .01$) relative to negative items ($M = .72$, $SE = .01$; $F(1,403) = 10.96$, $p = .001$, $\eta^2_p = .03$), but this did not differ as a function of age ($F(1,403) = 2.87$, $p = .09$, $\eta^2_p = .007$), manipulation ($F(2,403) = .13$, $p = .88$, $\eta^2_p = .001$), or the age-by-manipulation interaction ($F(2,403) = .16$, $p = .85$, $\eta^2_p = .001$). There was not a significant main effect of age ($F(1,403) = .003$, $p = .95$, $\eta^2_p < .001$) or manipulation ($F(2,403) = .99$, $p = .37$, $\eta^2_p = .005$), or an interaction of these factors ($F(2,403) = .72$, $p = .49$, $\eta^2_p = .004$; See Fig 2B). In an exploratory analysis, memory accuracy did not differ between the control condition and any manipulation condition, separately ($F(1,129) = .26$, $p = .61$, $\eta^2_p = .002$; $F(1,206) = .07$, $p = .93$, $\eta^2_p = .001$; $F(1,194) = 1.03$, $p = .36$, $\eta^2_p = .01$ for autobiographical, self, and music manipulations, respectively) or the timing of these manipulation conditions (see S2 File for analyses and Fig 3A for summary data).

**Individual difference analyses.** As in Experiment 1, we next considered different ways in which participants may engage with the autobiographical memory task. First, we compared participants in the control group ($N = 69$) to individuals in the autobiographical memory condition with *highly specific* autobiographical memories ($N = 32$) and to those with *less specific* autobiographical memories ($N = 32$). Replicating Experiment 1 findings, there was a significant effect of group ($F(1,127) = 5.01$, $p = .008$, $\eta^2_p = .07$), where participants with more specific memories ($M = .78$, $SE = .02$) had greater memory accuracy than those with less specific memories ($M = .67$, $SE = .02$, $p = .003$; See Fig 3B). There was also a trend for *high specificity* participants to perform better than those in the control group ($M = .73$, $SE = .02$, $p = .06$), and control participants did not differ from *low specificity* participants ($p = .1$). The effect of group did not interact with age or valence (all $p$s>.05).

Next, we compared participants in the control condition to those in the autobiographical memory condition who used a high proportion of positive emotional words in their narratives ($N = 32$), and those with a low proportion of positive words ($N = 32$). A second analysis compared control participants to those with high ($N = 32$) and low negative word use ($N = 33$). There was no effect of positive ($F(1,126) = .89$, $p = .41$, $\eta^2_p = .003$) or negative ($F(1,127) = .04$, $p = .97$, $\eta^2_p = .001$) word use on episodic memory performance. There was no interaction of positive or negative word use with age or valence (all $p$s>.05; See Fig 3C&3D). These findings replicated the benefit of specific autobiographical memories from Experiment 1, but showed that the benefit of negative autobiographical memory seen for neutral events in Experiment 1 did not generalize to emotional information in Experiment 2.

As with the autobiographical memory analysis, valence of self-statements did not influence emotional episodic memory performance ($F(1,206) = .09$, $p = .92$, $\eta^2_p = .001$ and $F(1,206) = .43$, $p = .66$, $\eta^2_p = .004$ for statement negativity and positivity, respectively). There were no interactions between production of positive or negative statements and age or event valence (all $p$s>.05; See Fig 3C&3D).

Individual differences in working memory, attention, community integration, crystalized knowledge, and sleep did not interact with the effect of manipulation on memory accuracy

(see S1 File for these analyses). In both models where we controlled for depression scores (i.e. GDS and BDI), there was a significant effect of valence on memory accuracy ($F(1,396) = 11.67$, $p = .001$, $\eta^2_p = .03$ and $F(1,396) = 11.59$, $p = .001$, $\eta^2_p = .03$ for models controlling for BDI and GDS, respectively). These were qualified by a significant valence-by-manipulation-by-GDS interaction (*F(1,396) = 2.92, p = .06, $\eta^2_p$ = .02*) and a trending valence-by-manipulation-by-BDI interaction ($F(1,396) = 3.71$, $p = .03$, $\eta^2_p = .02$). In both models, this was driven by a significant valence-by-depression interaction in individuals who had a socioemotional manipulation prior to encoding (*F(1,129) = 5.93, p = .02, $\eta^2_p$ = .04 and F(1,129) = 7.28, p = .008*, $\eta^2_p = .05$ for BDI and GDS, respectively). Although no particular correlations were significant, in individuals with greater depression scores, completing a socioemotional manipulation prior to encoding led to numerically reduced memory for negative items. All other contrasts were not significant.

Similarly, when controlling for self-reported loneliness, there were significant effects of valence ($F(1,395) = 10.96$, $p = .001$, $\eta^2_p = .03$), valence-by-age ($F(1,395) = 4.18$, $p = .04$, $\eta^2_p = .01$), valence-by-age-by-loneliness ($F(1,395) = 4.18$, $p = .04$, $\eta^2_p = .01$), and, critically, valence-by-manipulation-by-age-by-loneliness ($F(1,395) = 3.18$, $p = .04$, $\eta^2_p = .02$). Follow-up analyses revealed that loneliness was associated with better negative event memory in young adults ($r = .37$, $p = .08$) but not older adults ($r = -.04$, $p = .80$), in the control condition, only ($F(1,64) = 6.56$, $p = .01$, $\eta^2_p = .09$). All other contrasts were not significant.

Experiment 1 v. Experiment 2 comparison

A final analysis directly compared the effects of manipulation in Experiment 1 and Experiment 2. Participants in Experiment 2 (i.e., the *emotional image* memory task) had significantly greater memory accuracy ($M = .73$, $SE = .01$) compared to those in Experiment 2 (i.e., the *neutral image* memory task; $M = .67$, $SE = .01$; $F(1,2018) = 27.86$, $p < .001$, $\eta^2_p = .01$). There was also a significant positive effect of age on memory ($F(1,2018) = 14.36$, $p < .001$, $\eta^2_p = .007$), and an age-by-experiment interaction ($F(1,2018) = 14.92$, $p < .001$, $\eta^2_p = .007$), driven by greater age effects in Experiment 1 relative to Experiment 2 (Fig 2). There was not a significant effect of Manipulation ($F(1,2018) = 1.95$, $p = .14$, $\eta^2_p = .002$), and Manipulation did not interact with any other variables [Manipulation-by-experiment: $F(1,2018) = .07$, $p = .94$, $\eta^2_p < .001$; Manipulation-by-age: $F(1,2018) = .99$, $p = .37$, $\eta^2_p = .001$; Manipulation-by-experiment-by-age: $F(1,2018) = .21$, $p = .81$, $\eta^2_p < .001$].

**Summary.** As in Experiment 1, increased *specificity* in the AM condition was associated with greater memory accuracy in Experiment 2. Although the only significant difference between experiments was a greater positive effect of age on accuracy in Experiment 1, none of the other Experiment 1 findings replicated in Experiment 2. Unlike what was seen in Experiment 1, there was no indication that accuracy in Experiment 2 varied as a function of manipulation condition, and autobiographical memory and self-statement valence were not associated with memory accuracy.

Inclusion of emotional stimuli and affective scales in Experiment also allowed us to explore potential carry-over effects related to affective states. Surprisingly, the socioemotional manipulation prior to encoding may have buffered against negative memory biases for those with high depression scores; in these individuals, there was *reduced* negative memory accuracy for individuals who had completed a socioemotional manipulation prior to encoding, although nonsignificant simple effects make this effect difficult to interpret.

## General discussion

In the current study we had hoped to identify tasks that most older adults could complete prior to encoding or retrieval to boost their ability to remember information. Unfortunately,

the null results in these well-powered studies suggest that the manipulations we tested are unlikely to be useful in this way. There are two broad reasons why these null effects may have arisen, and they suggest different paths forward in terms of identifying pre-encoding and pre-retrieval strategies that may be more beneficial to older adults.

One explanation for the null effects in the current study is that the selected manipulations did not lead to an alteration in the underlying brain states and encoding or retrieval modes. These tasks were selected because they are associated with increased recruitment of the medial prefrontal cortex and other key midline structures relative to non-emotional and non-social cognitive tasks conducted in the scanner [19, 36, 37], but it is possible that differences in neural recruitment are a lot smaller when in the context of daily life. Further, the online study design may have made it more difficult for participants to fully engage in the task. Distractions and background tasks could have interfered with the ability of the manipulations to influence cognition in a way that they might have done in-person. A final possibility is that the manipulations may have affected the underlying brain state temporarily, but this effect faded too quickly to influence the encoding or retrieval modes engaged during the tasks. Future fMRI studies should be conducted to examine the effects of socioemotional tasks—such as the autobiographical memory, self reference, and music conditions tested here—on mPFC activation and connectivity. In particular, studies could examine changes to resting state functional connectivity between the mPFC and memory regions, such as the hippocampus, before and after socioemotional tasks.

A second possibility is that the selected manipulations may have altered the brain state and encoding or retrieval modes of participants as expected, but on average, that may convey no benefit to memory performance. In other words, although we know that older adults do better on memory tasks that rely on regions associated with socioemotional processing (i.e., mid-line regions such as the medial prefrontal cortex), priming these networks may not allow them to rely on these regions for unrelated memory tasks. Future studies could measure pre-task resting state connectivity between the mPFC and memory networks to establish individual differences in the tendency to rely on mPFC during memory tasks. A significant relation between mPFC connectivity and memory performance would suggest that the mPFC contributes to an ideal brain state for encoding and retrieval in older adults.

Finally, the results of the current study suggest that socioemotional manipulations do not convey an identical benefit across all circumstances. Results of Experiment 1 revealed a significant mnemonic benefit for the pre-retrieval music manipulation relative to pre-encoding music, suggesting a potential use of musical interventions to boost retrieval (but not encoding) modes in individuals with impaired memory. This finding should be interpreted cautiously, as it was an exploratory analysis that did not replicate when *emotional* memory was assessed rather than *neutral* memory (i.e., Expt. 2) and did not lead to an improvement over the control condition. In addition, the current study utilized a pleasant clip, but the results from the autobiographical memory and self statement conditions suggest that negative music might confer a greater benefit. Future research is needed to determine how features of the clip (e.g., valence, tempo, and familiarity) or characteristics of the participant (e.g., musical knowledge and experience) could boost this effect.

Neutral content was also enhanced in individuals who retrieved negative autobiographical events and self-reference statements, where emotional information was not. This finding could suggest a carry-over enhancing effect of negative emotion on the subsequent unrelated memory tasks, a benefit that would not necessarily benefit an emotional memory task that was already receiving an emotional memory enhancement. This perspective is supported by the similar episodic memory performance in these high negative Experiment 1 participants and all Experiment 2 participants. However, these findings could instead highlight an effect of

individual differences: individuals who are more inclined to retrieve negative or specific content in autobiographical or self-reference tasks may also have better memory in episodic memory tasks. A future study could explore these hypotheses by explicitly instructing participants to retrieve memories of different valences (e.g., positive v. negative) in separate retrieval blocks and measuring the effects on memory.

In both Experiments, episodic memory was greatest following the retrieval of specific autobiographical memories. As with the negativity analysis described above, it is difficult to interpret this pattern with the current design. On one hand, retrieval of a specific autobiographical event may induce a retrieval mode that supports better performance in the unrelated episodic memory task that follows. Alternatively, this analysis may have captured individual differences in memory ability (i.e., participants who can remember autobiographical events in more detail may naturally do better on episodic memory tasks). As described above, a future within-subject study could compare participants' memory after being instructed to retrieve a specific or general autobiographical memory.

Notably, none of the effects in the current study interacted with participant age, suggesting that these manipulations did not differentially affect young and older adults, as was expected. These null results are difficult to interpret, given the unexpected age-related increase in memory accuracy in Experiment 1, a finding that is counter to a majority of the cognitive aging literature (e.g., Park & Festini, 2017). Even in Experiment 2, where there was not a memory increase, we did not detect the expected age-related decrease in memory. The current study was designed to "rescue" impaired memory in older adults by reducing age-related deficits in memory. The lack of such deficits in the control condition means that the sample may not have been suited to test these hypotheses.

## Limitations

The online nature of the current research has some significant limitations that should be considered when interpreting the results. First, the results of the current study may be limited to older adults who have the skills and resources that allow them to easily access online research studies. This selection bias may be one reason why we saw unexpected age-related increases in memory accuracy. The current research was also affected by the onset of the COVID-19 pandemic toward the end of Experiment 1. The onset of the pandemic was associated with increases in anxiety and stress [60–62] which may have made it more difficult for participants to learn new information. Stay-at-home orders and changes to work-from-home policies may also have influenced the sample of participants available for online research studies. Exploratory analyses examining the effects of COVID timing suggest better performance in young, but not older, adults recruited after the onset of the pandemic (see S1 File) such that the unexpected age-related increases in accuracy were actually worse *before* the pandemic. COVID timing was not associated with socioemotional manipulations.

## Conclusions

A wealth of prior research has demonstrated the benefits of socioemotional processing to older adults' memory performance, but these studies have focused on processing *during* memory encoding or retrieval. The current study was the first to explore the benefits of socioemotional tasks conducted *prior to* memory in establishing an encoding or retrieval mode. Across two online studies, the current research fails to show evidence that socioemotional manipulations completed prior to encoding or retrieval reliably enhance memory, either for all participants or for older adults, more specifically. These null results suggest that socioemotional processing does not establish a supportive brain state that carries over to subsequent memory tasks but

future fMRI research should be conducted to better understand these patterns. In particular, these studies will be able to distinguish between two theoretically distinct explanations for these null results: 1) that these selected manipulations failed to alter participant brain states *or* 2) that the altered brain states failed to support better memory.

The current study also serves as an important reminder that individual differences should be considered when evaluating the effects of a manipulation. In particular, the current study revealed that how a participant engaged with a manipulation (e.g., whether they recalled a positive or negative memory) affected the strength of the manipulation. Future studies should be conducted using within-subject manipulations and explicit instructions that can help control for these individual differences.

## Supporting information

**S1 Fig. Age distributions as a function of experiment and condition.**
(DOCX)

**S1 Table. Education level as a function of experiment and condition.**
(DOCX)

**S2 Table. Race and ethnicity as a function of experiment and condition.**
(DOCX)

**S1 File. Additional individual difference analyses.** Experiment 1 COVID timing (pre- v. post- shut down orders) Attention (Test of Everyday Attention, TEA) Community Integration (Community Integration Questionnaire, CIQ) Working Memory (n-back: 2-back v. 0-back performance) Sleep (Pittsburg Sleep Quality Index, PSQI) Experiment 2 Attention (Test of Everyday Attention; TEA) Community Integration (Community Integration Questionnaire, CIQ) Working Memory (n-back: 2-back v. 0-back performance) Crystallized Knowledge (Shipley Vocabulary Test) Sleep (Pittsburg Sleep Quality Index, PSQI) Depression (Geriatric Depression Scale, GDS, and Beck's Depression Inventory, BDI) Loneliness (UCLA Loneliness Scale.
(DOCX)

**S2 File. Exploratory analyses examining separate effects of self, music, and autobiographical memory manipulations (Experiment 2).**
(DOCX)

**S3 File. Effects of age and manipulation on vividness rating.**
(DOCX)

**S4 File. Within-subjects analysis conducted on participants who completed longitudinal version of the study.**
(DOCX)

## Acknowledgments

The authors would like to thank Grace Serpe, Erin Davies, and Rachel Furlan helping to score behavioral data. All data (4 files) are available from the OSF database (DOI 10.17605/OSF.IO/3P6MA).

## Author Contributions

**Conceptualization:** Jaclyn H. Ford, Elizabeth Kensinger.

**Data curation:** Jaclyn H. Ford, Ryan Daley, Julia Maybury, Cortney Stedman, Julia Swiatek, Rachel Van Boxtel, Erin Welch.

**Formal analysis:** Jaclyn H. Ford, Julia Maybury, Cortney Stedman, Julia Swiatek, Rachel Van Boxtel, Erin Welch.

**Funding acquisition:** Jaclyn H. Ford, Elizabeth Kensinger.

**Investigation:** Jaclyn H. Ford.

**Methodology:** Jaclyn H. Ford, Ryan Daley, Cortney Stedman, Julia Swiatek, Rachel Van Boxtel, Erin Welch, Elizabeth Kensinger.

**Project administration:** Jaclyn H. Ford, Elizabeth Kensinger.

**Supervision:** Jaclyn H. Ford, Elizabeth Kensinger.

**Visualization:** Jaclyn H. Ford, Ryan Daley.

**Writing – original draft:** Jaclyn H. Ford.

**Writing – review & editing:** Jaclyn H. Ford, Ryan Daley, Julia Maybury, Cortney Stedman, Julia Swiatek, Rachel Van Boxtel, Erin Welch, Elizabeth Kensinger.

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
