## [Decision Letter · Decision Letter 0]

18 Jun 2024

PONE-D-24-15530Limited carry-over effects of socioemotional manipulations on subsequent unrelated memory tasksPLOS ONE

Dear Dr. Ford,

Thank you for submitting your manuscript to PLOS ONE. After careful consideration, we feel that it has merit but does not fully meet PLOS ONE’s publication criteria as it currently stands. Therefore, we invite you to submit a revised version of the manuscript that addresses the points raised during the review process.

**ACADEMIC EDITOR: **The decision for a major revision was made based on the reviewers' comments. Both reviewers believe the article has the potential for publication with appropriate revisions. They have provided recommendations to enhance the article's solidity. Please proceed with these revisions to improve your article and meet the publication criteria.

We look forward to receiving your revised manuscript.

Kind regards,

João Carlos Gonçalves dos Reis

Academic Editor

PLOS ONE

 [National Science Foundation (BCS-1823795) to EAK and JHF].  

[ This research was supported by a grant from the National Science Foundation (BCS-1823795) to EAK and JHF. The authors would like to thank Grace Serpe, Erin Davies, and Rachel Furlan helping to score behavioral data. All data (4 files) are available from the OSF database (DOI 10.17605/OSF.IO/3P6MA).]

 [National Science Foundation (BCS-1823795) to EAK and JHF].

Additional Editor Comments:

The decision for a major revision was made based on the reviewers' comments. Both reviewers believe the article has the potential for publication with appropriate revisions. They have provided recommendations to enhance the article's solidity. Please proceed with these revisions to improve your article and meet the publication criteria.

Reviewers' comments:

Reviewer's Responses to Questions

**Comments to the Author**

1. Is the manuscript technically sound, and do the data support the conclusions?

Reviewer #1: Yes

Reviewer #2: Yes

2. Has the statistical analysis been performed appropriately and rigorously? 

Reviewer #1: Yes

Reviewer #2: Yes

3. Have the authors made all data underlying the findings in their manuscript fully available?

Reviewer #1: Yes

Reviewer #2: Yes

4. Is the manuscript presented in an intelligible fashion and written in standard English?

Reviewer #1: Yes

Reviewer #2: Yes

5. Review Comments to the Author

Reviewer #1: The article "Limited Carry-Over Effects of Socioemotional Manipulations on Subsequent Unrelated Memory Tasks" explores the hypothesis that performing socioemotional tasks before encoding or retrieval can improve performance in subsequent unrelated memory tasks. Conducted with a large and diverse sample of adults, the study includes two online experiments. In the first experiment, participants completed memory tasks with pairs of neutral words and images after being subjected to different socioemotional manipulations (self-reference, music, or autobiographical memory retrieval) or a control condition. In the second experiment, the procedure was similar but used pairs of emotional words and images (positive or negative). The results showed that socioemotional manipulations did not significantly improve subsequent memory, suggesting that these interventions may not be effective in enhancing memory in unrelated tasks. However, exploratory analyses indicated that certain individuals, such as those who elaborated on specific autobiographical memories, might benefit from these manipulations, highlighting the need for future research to investigate specific conditions and individual characteristics that might moderate these effects.

I think that after an improvement of the article, it may be accepted for publication. I left my suggestions below.

1. In the introduction, there could be a clearer connection between the introduction and the specific objectives of the study.

2. The introduction could be more direct in pointing out the specific gap in the literature that the study addresses. For example, explicitly state how the current research goes beyond previous studies on socioemotional memory in the elderly.

3. In the "Methods" section, the description of the participants is detailed. However, it would be useful to include more information about age distribution and other important demographic characteristics, if possible.

4. Include more details about the specific procedures of the socioemotional manipulations. For example, describe exactly how participants were instructed to perform the self-reference and autobiographical memory tasks.

5. It would be helpful to add a figure or flowchart to illustrate the flow of the experiment.

6. The results are presented clearly but could benefit from additional tables and figures to visualize the data.

7. The exploratory analyses are well addressed, but there are many details. A more concise synthesis of the main findings could help with readability.

8. The limitations of the study are discussed, but it would be useful to add more suggestions for future research.

9. The presented figures should be of better quality.

10. More recent bibliographic references should be added.

11. An expanded version of the conclusion should be made.

Reviewer #2: I found the article entitled “Limited carry-over effects of socioemotional manipulations on subsequent unrelated memory tasks” very interesting. The article is interesting because it deals with a timely topic and because it presents some relevant results.

The authors adhered to the IMRAD structure, which is appropriate for a PLOS ONE article; however, the introduction heading is missing.

The conclusion is too brief. I suggest focusing on four main points to enhance it. First, highlight the contributions to the theory, specifying the novel and original insights your article adds. Second, discuss the practical contributions, detailing how your findings are relevant to practitioners in the field. Third, address the limitations of your research (this is already mentioned, but it should be included in the conclusion). Finally, provide suggestions for future research. The conclusion currently feels imbalanced compared to the other sections. These are only suggestions, and you can make adjustments as long as you expand the conclusion and emphasize the research contributions.

From a theoretical perspective, the article presents limitations that need a thorough revision. Specifically, the references include only sources from 2022 onwards, with only one article from 2022. It is essential to include articles from 2023 and 2024 that are currently missing. Resolving this issue is necessary for the article to be published.

Although I agree that the authors have done an excellent job, there is a lack of theoretical support. At the end of the article, the authors should focus on the contributions, as this is where readers will concentrate the most.

6. PLOS authors have the option to publish the peer review history of their article (what does this mean?). If published, this will include your full peer review and any attached files.

Reviewer #1: No

Reviewer #2: No

---

## [Author Response · Author response to Decision Letter 0]

22 Jul 2024

July 16, 2024 

Emily Chenette

Editor-in-Chief, PLOS ONE

Dear Dr. Chenette,

Thank you for reviewing our manuscript entitled, “Limited carry-over effects of socioemotional manipulations on subsequent unrelated memory tasks“. We sincerely appreciated the suggestions provided by the academic editor and 2 reviewers, as well as their positive response to the manuscript. 

As stated in the cover letter, the revised manuscript has been edited to conform with PLOS ONE’s stye requirements. The editor also requested an edit to the Role of Funder statement, stating that: “The funders had no role in study design, data collection and analysis, decision to publish, or preparation of the manuscript.” We have addressed the reviewers’ comments below and have incorporated these changes into the manuscript (in red). We hope that you will now find this manuscript suitable for publication in PLOS ONE.

Sincerely,

Jaclyn Hennessey Ford

Reviewer #1

1. The reviewer requested a clearer connection between the introduction and the specific objectives of the study.

Response: The introduction has been updated to include more connections between the literature and the current research questions. 

2. The reviewer noted that the introduction could be more direct in pointing out the specific gap in the literature that the study addresses. For example, they asked for an explicit statement as to how the current research goes beyond previous studies on socioemotional memory in the elderly.

Response: The current study is the first to consider the use of socioemotional tasks as a manipulation to establish an ideal encoding or retrieval mode for older adults. Other research has shown a benefit of socioemotional processing during memory tasks, but not prior to memory. This information has been highlighted in the introduction

3. The reviewer requested more information about the age distribution of participants.

Response: We have added violin plots of the age distribution as a function of condition and experiment. 

4. The reviewer requested more information about the instructions that participants were given for the socioemotional manipulations. 

Response: We have added the exact instructions provided to participants to the methods section of the manuscript. 

5. The reviewer requested a figure or flowchart to illustrate the flow of the experiment

Response: We have updated Fig 1 to better reflect the flow of the experiment. 

6. The reviewer requested additional tables and figures to visualize the data.

Response: The manuscript includes a table or figure for all reported findings, so we were not sure what other visualization the reviewer was hoping for. However, we would be happy to add an additional figure or table if the review has a specific request. 

7. The reviewer requested a concise synthesis of the main findings of the exploratory analyses. 

Response: The results section has been updated to include summaries of the findings in each paragraph. 

8. The reviewer requested additional suggestions for future research

Response: Suggestions for future research have been added throughout the discussion section

9. The reviewer noted that the figures should be saved at a higher quality

Response: The figures have been saved at a higher quality 

10. The reviewer noted a lack of recent (2023 and 2024) references 

Response: We have added a number of new references to keep the manuscript up-to-date.

11. The reviewer requested an expanded version of the conclusions paragraph 

Response: The conclusions paragraph now includes a more thorough discussion of the theoretical and methodological contributions of the study, as well as a review of future research. 

Reviewer #2

1. The reviewer noted that the introduction heading is missing

Response: This heading has been added 

2. The reviewer suggested focusing on four additional points to enhance the final conclusion paragraph, as it felt short and out of balance with the rest of the manuscript: 

a. First, they asked how the current findings might contribute to overarching theory. 

b. Second, they requested a discussion of the practical contributions of the study to practitioners in the field

c. Third, the reviewer asked that the limitations, discussed elsewhere in the discussion, be mentioned in the conclusion.

d. Finally, they request suggestions for future research

Response: The conclusions paragraph has been updated according to these suggestions. 

3. The reviewer noted that the manuscript was missing references from the past two years 

Response: We have added a number of new references to keep the manuscript up-to-date.

4. The reviewer requested additional focus on the theoretical contributions of the study at the end of the manuscript 

Response: A description of the theoretical contributions has been added as part of the updated conclusions

---

## [Decision Letter · Decision Letter 1]

7 Aug 2024

Limited carry-over effects of socioemotional manipulations on subsequent unrelated memory tasks

PONE-D-24-15530R1

Dear Dr. Ford,

We’re pleased to inform you that your manuscript has been judged scientifically suitable for publication and will be formally accepted for publication once it meets all outstanding technical requirements.

Kind regards,

Chongzeng Bi

Academic Editor

PLOS ONE

Additional Editor Comments (optional):

Reviewers' comments:

Reviewer's Responses to Questions

**Comments to the Author**

1. If the authors have adequately addressed your comments raised in a previous round of review and you feel that this manuscript is now acceptable for publication, you may indicate that here to bypass the “Comments to the Author” section, enter your conflict of interest statement in the “Confidential to Editor” section, and submit your "Accept" recommendation.

Reviewer #1: All comments have been addressed

Reviewer #2: All comments have been addressed

2. Is the manuscript technically sound, and do the data support the conclusions?

Reviewer #1: Yes

Reviewer #2: Yes

3. Has the statistical analysis been performed appropriately and rigorously? 

Reviewer #1: Yes

Reviewer #2: Yes

4. Have the authors made all data underlying the findings in their manuscript fully available?

Reviewer #1: Yes

Reviewer #2: Yes

5. Is the manuscript presented in an intelligible fashion and written in standard English?

Reviewer #1: Yes

Reviewer #2: Yes

6. Review Comments to the Author

Reviewer #1: I would like to thank the authors for effectively addressing the comments and suggestions made during the first round of review. After a careful analysis of the revisions, I am pleased to see that the authors have addressed the raised concerns, significantly improving the manuscript.

The authors have made the necessary revisions and I recommend that the article be accepted for publication, considering that the requested revisions have been satisfactorily incorporated and the manuscript now presents robust and well-founded content.

Reviewer #2: I would like to thank you for your hard work and effort in addressing all my comments. Congratulations on meeting the rigorous standards of PLOS ONE!

7. PLOS authors have the option to publish the peer review history of their article (what does this mean?). If published, this will include your full peer review and any attached files.

Reviewer #1: No

Reviewer #2: **Yes: **João Carlos Gonçalves dos Reis

---

## [Editor Report · Acceptance letter]

12 Aug 2024

PONE-D-24-15530R1 

PLOS ONE

Dear Dr. Ford, 

I'm pleased to inform you that your manuscript has been deemed suitable for publication in PLOS ONE. Congratulations! Your manuscript is now being handed over to our production team.

Kind regards, 

on behalf of

Professor Chongzeng Bi 

Academic Editor

PLOS ONE